# Repurposing of Fluvastatin as an Anticancer Agent against Breast Cancer Stem Cells via Encapsulation in a Hyaluronan-Conjugated Liposome

**DOI:** 10.3390/pharmaceutics12121133

**Published:** 2020-11-24

**Authors:** Ji Su Yu, Dae Hwan Shin, Jin-Seok Kim

**Affiliations:** 1Research Institute of Pharmaceutical Science, College of Pharmacy, Sookmyung Women’s University, Cheongparo 47 gil 100, Yongsan gu, Seoul 04310, Korea; yujs0910@naver.com; 2College of Pharmacy, Chungbuk National University, Cheongju 28160, Korea; dshin@chungbuk.ac.kr

**Keywords:** fluvastatin, breast cancer stem cells, hyaluronan, drug delivery system, liposome

## Abstract

Fluvastatin (FLUVA), which is a common anti-hypercholesterolemia drug, exhibits potential anticancer activity as it suppresses the proliferation, angiogenesis, and metastasis of breast cancer cells via inhibiting 3-hydroxy-methyl glutaryl-coenzyme A (HMG-CoA) reductase. In this study, hyaluronan-conjugated FLUVA-encapsulating liposomes (HA-L-FLUVA) were evaluated for their anticancer efficacy in vitro and in vivo. The particle size, zeta potential, and encapsulation efficiency of HA-L-FLUVA were 158.36 ± 1.78 nm, −24.85 ± 6.26 mV, and 35%, respectively. Growth inhibition of breast cancer stem cells (BCSCs) by HA-L-FLUVA was more effective than that by free FLUVA. The half maximal inhibitory concentration (IC_50_) values of FLUVA, L-FLVUA, and HA-L-FLUVA were 0.16, 0.17, and 0.09 μM, respectively. The in vivo anticancer effect of HA-L-FLUVA in combination with doxorubicin (DOX) was more effective than that of free FLUVA, free DOX, and HA-L-FLUVA. The longest survival of mice was achieved by treatment with FLUVA (15 mg/kg) and HA-L-FLUVA (15 mg/kg) + DOX (3 mg/kg), followed by HA-L-FLUVA (15 mg/kg), Dulbecco’s phosphate buffered saline, and DOX (3 mg/kg). No more than 10% body weight loss was observed in the mice injected with FLUVA, indicating that the drug was not toxic. Taken together, these results indicate that HA-L-FLUVA could serve as an effective anticancer drug by inhibiting the growth of both breast cancer cells and cancer stem cells.

## 1. Introduction

Breast cancer is one of the most frequently diagnosed cancers in women and the leading cause of cancer-related deaths worldwide [1,2]. Breast cancer is a malignant tumor with invasive and metastatic abilities. The main cellular burden in breast cancer consists of the so-called bulk tumor cells and cancer stem cells (CSCs), which constitute a small percentage of the tumor bulk. CSCs possess the characteristics of both stem cells and cancer cells as they have the self-renewal capability of normal stem cells, as well as the independent growth, tumorigenicity, and metastatic potential of cancer cells [3]. In addition, they are resistant to conventional therapies, such as chemotherapy and radiotherapy [4,5]. The presence of a small population of breast CSCs (BCSCs) is recognized as one of the causes of breast cancer recurrence [6,7]. CSCs are characterized by a specific surface marker phenotype, and BCSCs have been identified to have the CD44^+^/CD24^−/low^ phenotype [8,9]. Preclinical and clinical trials have attempted to establish novel therapeutic regimens that aim to eradicate CSCs for the complete treatment of cancer [10,11,12]. As CD44 is known to be overexpressed on BCSCs, surface marker-targeting ligand-conjugated drug delivery systems, such as liposomes, could be used to target BCSCs and enhance the overall therapeutic efficacy of drugs against breast cancer [13,14,15].

Fluvastatin (FLUVA), which is also known as a 3-hydroxy-methyl glutaryl-coenzyme A (HMG-CoA) reductase inhibitor, belongs to the class of lipid-lowering drugs that block the conversion of HMG-CoA to mevalonic acid [16,17,18,19]. Recent studies have reported that FLUVA is a novel therapeutic agent for breast cancer prevention and treatment [17]. FLUVA belongs to the class of drugs that inhibit the mevalonate pathway, which is essential for the synthesis of compounds that significantly affect several critical cellular functions, such as cell membrane integrity, cell signaling, and cell cycle progression. These functions are related to cancer initiation, growth, and metastasis [20,21,22]. Therefore, FLUVA inhibits the proliferation and angiogenic and metastatic properties of breast cancer cells and BCSCs [23]. However, FLUVA has a low oral bioavailability (~24%), short half-life, and 98% plasma protein binding. Therefore, a higher amount of FLUVA is required for cancer treatment than for treating hypercholesterolemia, which may give rise to the typical adverse effects of FLUVA, including hepatotoxicity, peripheral neuropathy, and rhabdomyolysis [24].

Hyaluronan (HA), which is also known as hyaluronic acid or hyaluronate, is a major extracellular matrix component [25,26]. It is synthesized as a large, negatively charged, unbranched polymer composed of repeating disaccharides of glucuronic acid and *N*-acetylglucosamin [27]. It is a biodegradable and non-immunogenic biomaterial [28]. HA binds to CD44 on the cell surface, allowing the targeting of BCSCs that overexpress CD44 [29,30,31]. HA has also been reported to form a hydrophilic barrier and prolong the circulation time of HA-conjugated nanocarriers in blood [32].

Liposomes are the most common nanocarriers used for targeted drug delivery systems [33,34,35]. They have several advantages in overcoming obstacles to cellular uptake and improving the payload biodistribution [36]. Most solid tumors have unique pathological properties, such as extensive angiogenesis, defective vascular structures, and damaged lymphatic drainage systems [37]. Therefore, liposomes with a size ranging from 100 to 200 nm in diameter can easily penetrate the tumor and be retained for a prolonged period owing to the enhanced permeability and retention (EPR) effect [38]. In addition, the use of hydrophilic polymers, such as HA, to conjugate drugs to the liposome surface can prevent the uptake of drug-carrying liposomes by the reticuloendothelial system (RES) [39]. Moreover, targeting ligand-conjugated liposomes are delivered to the target site more effectively than the conventional non-conjugated liposomes [40]. 

In this study, HA-conjugated FLUVA-encapsulating (HA-L-FLUVA) liposomes were prepared and tested for their in vitro and in vivo anticancer effects using MCF-7-derived BCSCs and xenograft mouse models, respectively (Scheme 1). 

## 2. Materials and Methods 

### 2.1. Materials and Reagent

FLUVA, HA (1500 kDa), 1-ethyl-3-(3-dimethyl-aminopropyl-carbodiimide) (EDC), poly-2-hydroxyethyl methacrylate (poly-HEMA), cholesterol, bovine serum albumin (BSA), basic fibroblast growth factor (bFGF), human epidermal growth factor (hEGF), insulin solution, dimethyl sulfoxide (DMSO), accutase, boric acid, paraformaldehyde, ethanol, methanol, and chloroform were purchased from Sigma-Aldrich (St. Louis, MO, USA). l-α-phosphatidylcholine from egg (EPC) and 1,2-dioleoyl-sn-glycero-3-phosphonethanolamine (DOPE) were purchased from Avanti Polar Lipids (Alabaster, AL, USA). The MCF-7 human breast cancer cell line was purchased from the Korean Cell Line Bank (Seoul, Korea). RPMI-1640 (with l-glutamine and 25 mM HEPES), penicillin/streptomycin, Dulbecco’s phosphate buffered saline (DPBS), and Trypsin-EDTA were purchased from Welgene, Inc. (Daegu, Korea). Fetal bovine serum (FBS), DMEM/F12, and B-27 supplement were purchased from Gibco (Grand Island, NY, USA). Anti-CD44-phycoerythrin (PE) and anti-CD24-fluorescein isothiocyanate (FITC) antibodies were purchased from Miltenyi Biotec (Bergisch Gladbach, Germany) and 17 beta-estradiol pellets (0.18 mg/60 days) were purchased from Innovative Research of America (Sarasota, FL, USA). 

### 2.2. Methods

#### 2.2.1. Preparation of Liposomal FLUVA (L-FLUVA) and HA-L-FLUVA

L-FLUVA was prepared by a thin lipid film hydration technique followed by freeze-thawing. L-FLUVA was composed of EPC, cholesterol, and DOPE (55:45:20 molar ratio). First, 52.8 μmol of lipid mixture in chloroform and FLUVA in methanol were mixed in a round-bottomed flask. The organic solvent in the mixture was evaporated using a rotary evaporator (Laborota 4000; Heidolph Instrument, Schwabach, Germany) under 6 min of exposure to nitrogen gas flow and then further evaporated for 45 min in a 42 °C water bath to form a thin lipid film. The lipid film was then hydrated using 10 mM HEPES buffer (pH 7.5). After hydration, the liposomes were subjected to 25 cycles of freezing (liquid nitrogen) and thawing (42 °C water bath) to produce smaller unilamellar liposomes with an enhanced encapsulation efficiency. L-FLUVA was extruded using a mini-extruder (Avanti^®^ polar lipid, Inc., Alabaster, AL, USA) through 800-, 400-, and 200-nm polycarbonate membrane filters (Whatman International, Ltd., Maidstone, UK), three times for each filter. 

EDC coupling for amide synthesis was conducted to conjugate the HA onto the DOPE, which is lipid on the surface of L-FLUVA [41,42]. Briefly, 4 mg of HA was dissolved in 1 mL of distilled water and mixed with 2 mg of EDC. The mixture was activated at pH 4 for 2 h at 37 °C. After activation, the HA solution was mixed with L-FLUVA (1:1 molar ratio of HA:liposome). The pH of the mixture was adjusted to 8.6 with 0.1 M borate buffer (pH 9.4). The reaction was performed for 24 h at 37 °C. Unencapsulated FLUVA and unconjugated components were removed from the liposome solution by Sepharose gel column chromatography and ultracentrifugation (20,000× *g*, 4 °C, 40 min).

#### 2.2.2. Analysis of the Size Distribution and Zeta Potential of the Liposome 

The size distribution and zeta (ζ)-potential of liposomes were measured by dynamic laser-light scattering (SZ-100; Horiba, Japan). Each FLUVA liposomal formulation was diluted in distilled water at a 1:100 volume ratio and analyzed at 20–22 °C with a scattering angle of 90°.

#### 2.2.3. Drug Encapsulation Efficiency

FLUVA encapsulated in liposomes was extracted by the Bligh and Dyer extraction method [43]. Briefly, 100 μL of liposome was mixed with 1 mL of chloroform, 250 μL of methanol, and 150 μL of DPBS. The mixture was vortexed until homogeneous. Next, it was centrifuged at 2700× *g* for 15 min at 20–22 °C, resulting in separation into two phases. The lower (organic) phase contained lipophilic materials, including FLUVA in chloroform, and the upper (aqueous) phase contained hydrophilic materials in methanol. Next, 1 mL of chloroform was added to the lower organic phase, and the mixture was centrifuged again. These steps were repeated three times. The amount of FLUVA in the organic phase was measured at 304 nm using a UV spectrophotometer (Ultraspec 4000; Pharmacia biotech, Piscataway, NJ, USA). The FLUVA encapsulation efficiency (%) of the liposomes was calculated using the following formula:$$\mathrm{FLUVA}\;\mathrm{encapsulation}\;\mathrm{efficiency}\;\left(\%\right)=\left(\frac{\mathrm{Amount}\;\mathrm{of}\;\mathrm{FLUVA}\;\mathrm{in}\;\mathrm{liposome}}{\mathrm{Initial}\;\mathrm{amount}\;\mathrm{of}\;\mathrm{FLUVA}}\right)\times 100.$$

#### 2.2.4. Culture of 2D Bulk Cells (MCF-7 Adherent Cells) and BCSCs (MCF-7 Non-Adherent Cells)

The 2D bulk cells were cultured as monolayers in RPMI-1640 medium containing 10% FBS and 100 unit/mL penicillin/streptomycin at 37 °C in a 5% CO_2_ incubator (Sanyo Electric Co., Ltd., Moriguchi City, Osaka, Japan). When the cells reached approximately 80% confluence, they were washed with DPBS, trypsinized for 2 min at 37 °C, and centrifuged at 125× *g* for 3 min. Next, the cells were washed with serum-free medium containing 100 unit/mL penicillin/streptomycin, centrifuged again, and cultured as mentioned earlier. 

For the non-adherent BCSC culture, the cultured 2D bulk cells were dissociated using Trypsin-EDTA, and single cells were cultured on poly-HEMA-coated culture dishes containing fresh culture medium. The BCSCs were cultured in serum-free medium containing 100 unit/mL penicillin/streptomycin, 0.4% BSA, 5 μg/mL insulin, 20 ng/mL bFGF, 20 ng/mL hEGF, and B-27 supplement in DMEM/F-12. Under these conditions, the 2D bulk cells grew as non-adherent spheres, called mammospheres. Fresh culture medium was added on the fourth day. The BCSCs were collected on the seventh day and dissociated into single cells using accutase (10 min at 37 °C).

#### 2.2.5. Flow Cytometric Analysis of BCSCs 

The BCSCs and 2D bulk cells were dissociated into single cells and washed twice with DPBS. For flow cytometric analysis, the single cells of BCSCs and 2D bulk cells were stained in the dark with anti-CD44-FITC and anti-CD24-PE antibodies in 100 μL DPBS for 30 min at 4 °C. The stained cells were washed and resuspended in cold DPBS. CD44^+^/CD24^−/low^ surface markers were determined using a FACSCalibur™ flow cytometer (BD Bioscience, San Jose, CA, USA).

#### 2.2.6. Inhibition of BCSC Proliferation

The anti-proliferative effect of FLUVA formulations in BCSCs was evaluated using a sphere-forming assay. Single cells of BCSCs were seeded in poly-HEMA-coated 96-well plates at a density of 2000 cells/well in 100 μL of serum-free medium. The cells in each well were treated with free FLUVA or HA-L-FLUVA in the medium. The plates were incubated at 37 °C in a 5% CO_2_ incubator. After 7 days, the number of spheres with a diameter of ≥100 μm was counted under a microscope (ix71; Olympus, Tokyo, Japan). To compare the inhibition of the sphere-forming ability, the half maximal inhibitory concentration (IC_50_) was calculated using the following GraphPad Prism 9.0 (Prism Software, Inc., San Diego, CA, USA) formula:$${\mathrm{IC}}_{50}=\frac{\mathrm{sphere}\;\mathrm{number}\;\mathrm{of}\;\mathrm{sample}}{\mathrm{sphere}\;\mathrm{number}\;\mathrm{of}\;\mathrm{control}}\times 100.$$

#### 2.2.7. Antitumor Effects of HA-L-FLUVA in Xenograft Mouse Models

All mouse experiments were approved by the Institutional Animal Care and Use Committee of Sookmyung Women’s University (SMU-IACUC), Korea. BALB/C nude mice (female, 5 weeks old) were purchased from Nara-Biotec (Seoul, Korea). All experiments were performed in accordance with SMU-IACUC guidelines and procedures (approval number: SMWU-IACUC-1906-015, 24 October 2019). The antitumor effect of HA-L-FLUVA was evaluated in a BCSC-xenografted mouse model. The mice were housed in ventilated cages with free access to water and food. 17β-estradiol pellets (0.18 mg/60 days) were implanted subcutaneously (s.c.) around the neck to establish xenograft mouse models [44,45]. After 3 days, the BCSCs were harvested from the culture dishes and washed three times with DPBS. Next, BCSCs (1 × 10^6^) in 40 μL DPBS and 60 μL Matrigel were injected s.c. into the right flank of the mice. Within 10 days, all mice developed tumors with a size of approximately 60–70 mm^3^. The mice were randomly distributed into five groups (n = 7 per group). The mice in each group were injected retro-orbitally with DPBS (control), FLUVA (15 mg/kg), doxorubicin (DOX) (3 mg/kg), HA-L-FLUVA (15 mg/kg), or HA-L-FLUVA with DOX (HA-L-FLUVA (15 mg/kg) + DOX (3 mg/kg)). The mice were injected with the respective FLUVA formulation every day for 2 weeks and/or DOX every 4 days for 2 weeks. The tumor size was measured using a Vernier caliper at 2- or 3-day intervals and calculated using the standard formula: $$\mathrm{Tumor}\;\mathrm{size}=\frac{{\left(\mathrm{width}\right)}^{2}\times \mathrm{length}}{2}.$$

#### 2.2.8. In Vivo Toxicity Determination

The mice in each group were injected retro-orbitally with DPBS (control), FLUVA (15 mg/kg), DOX (3 mg/kg), HA-L-FLUVA (15 mg/kg), or HA-L-FLUVA (15 mg/kg) + DOX (3 mg/kg). FLUVA formulations were injected every day for 2 weeks and DOX was injected every 4 days for 2 weeks. The body weight changes of all groups were measured at 2- or 3-day intervals. The toxicity of drugs was defined as >10% body weight loss, abnormal behavior, signs of discomfort, or the death of animals [46]. Normalization of body weight change was calculated by dividing the body weight by the initial body weight.

#### 2.2.9. Statistical Analysis

All results are presented as the mean ± standard deviation (SD). Multiple comparisons were conducted using two-way analysis of variance (ANOVA) with Turkey’s post-hoc test for BCSC anti-proliferation data. In vivo parameters were performed using one-way ANOVA with Turkey’s post-hoc test. A *p*-value of less than 0.05 was considered statistically significant.

## 3. Results

### 3.1. Analysis of the Size Distribution, Zeta Potential, and FLUVA Encapsulation Efficiency (%) of Liposomes 

The mean diameters of liposomes were 115.7 ± 2.00 and 158.4 ± 1.78 nm for L-FLUVA and HA-L-FLUVA, respectively (Table 1). A proper particle size ranging from 100 to 200 nm is known to improve the biodistribution via accumulation in tumor tissues (the EPR effect) [38]. The zeta potential was −8.13 ± 7.72 and −24.85 ± 6.26 mV for L-FLUVA and HA-L-FLUVA, respectively.

The encapsulation efficiency was analyzed using UV spectrophotometry by detecting the absorbance of FLUVA at 304 nm. The FLUVA encapsulation efficiency of L-FLUVA and HA-L-FLUVA was approximately 33% and 35%, respectively.

### 3.2. Cell Culture of 2D Bulk Cells and BCSCs 

2D bulk cells adhered to the surface of the culture dish and grew. BCSCs grew as non-adherent mammospheres in poly-HEMA-coated culture dishes (Figure 1A). To characterize the phenotypes of 2D bulk cells and BCSCs, cells were stained with 5 μL anti-CD44-FITC and 5 μL anti-CD24-PE. The population of BCSCs was identified as CD44^+^/CD24^−/low^. The percentage of BCSCs in mammospheres was 26.93%, whereas that of 2D bulk cells was 2.05% (Figure 1B).

### 3.3. Inhibition of BCSC Proliferation

To investigate the anti-proliferative effects of FLUVA in various formulations on BCSCs, we treated BCSCs with different FLUVA formulations for 7 days. The number and size of the spheres decreased when the FLUVA concentration was increased (Figure 2). The IC_50_ values of FLUVA, L-FLVUA, and HA-L-FLUVA were 0.16, 0.17, and 0.09 μM, respectively (Figure 3 and Table 2). The inhibitory effect of HA-L-FLUVA on sphere formation was approximately two-fold higher than FLUVA or L-FLUVA. There was no distinctive difference between FLUVA and L-FLVUA in this experiment.

### 3.4. Antitumor Effects of HA-L-FLUVA in BCSC Xenograft Mouse Models 

The antitumor effect of HA-L-FLUVA was assessed in a BCSC-xenografted mouse model (BALB/C nude mice). HA-L-FLUVA displayed a more potent anticancer effect than free FLUVA and DOX (Figure 4). Furthermore, the DOX-treated group did not show effective anticancer effects. However, the mice co-treated with HA-L-FLUVA and DOX exhibited the smallest increase in tumor size among the five treatment groups and more prolonged remission than the mice subjected to the other drug treatments (Figure 4). Moreover, 100% survival of mice was achieved by treatment with FLUVA (15 mg/kg) and HA-L-FLUVA (15 mg/kg) + DOX (3 mg/kg), whereas treatment with HA-L-FLUVA (15 mg/kg) and DPBS led to 85.7% survival. Furthermore, DOX (3 mg/kg) led to the lowest survival rate of 71.4%, as shown in Figure 5.

### 3.5. Determination of the In Vivo Toxicity 

A decrease in mouse body weight after treatment with drugs is related to the toxicity of the drugs per se or the delivery system, such as liposomes. The toxicity of drugs was defined as a >10% decrease in body weight. None of the drug-treated mice showed over a 10% body weight decrease during the 30-day period, which implies the absence of gross toxicity of the drug formulations used in the study (Figure 6).

## 4. Discussion

Recent studies state that most types of cancers may include CSCs, which are the main cause of recurrence and metastasis. One therapeutic strategy for breast cancer treatment is targeting the CD44 surface marker on BCSCs using HA.

Drug repurposing or drug repositioning is a strategy for identifying new uses of approved or investigational drugs that are outside the scope of the original medical indication [47]. FLUVA is a potent inhibitor of HMG-CoA reductase that can inhibit cholesterol synthesis or the isoprenoid pathway. Therefore, FLUVA can be used as an effective therapeutic agent for the prevention and treatment of breast cancer. However, the oral administration of FLUVA may cause some side effects, even though these side effects are less severe than the serious side effects of conventional anticancer agents. Therefore, we developed HA-L-FLUVA as an efficient anticancer agent that specifically targets BCSCs. 

The particle size of HA-L-FLUVA was larger than that of L-FLUVA, which may be caused by the addition of HA on the surface of the liposome. The EPR effect is related to the particle size. Specific-sized liposomes of approximately 100–200 nm allow molecules to accumulate at tumor sites at higher concentrations than at normal sites and be retained for a long time [38]. The measured zeta potentials of L-FLUVA and HA-L-FLUVA were in the range of 0 to −31 mV. Nanoparticles with a negative charge are much more stable and less toxic following an intravenous injection than particles with a positive or neutral charge. Therefore, we designed liposomes with suitable physicochemical characteristics for targeting BCSCs.

A subpopulation (CD44^+^/CD24^−/low^) of breast cancer cells has been reported to have stem cell properties [8,9]. To perform the targeting study, we identified the ratio of BCSCs and the 2D bulk cell population with the CD44^+^/CD24^−/low^ phenotype. The percentage of BCSCs was 13-fold higher than that of 2D bulk cells (Figure 1). This result is consistent with that of previous studies that cultured mammospheres that had a high percentage of CD44^+^/CD24^−/low^ [48,49].

Suppression of the BCSC characteristics by FLUVA formulations was identified through a sphere-forming assay. The sphere formation assay was first designed more than 25 years ago to separate neural stem cells [50]. This method is widely used for assessing the stemness and enrichment of CSCs. This assay has been applied for the generation and maintenance of CSCs with a higher tumorigenicity [51]. In this study, the number and size of the spheres were decreased following treatment with an increased concentration of FLUVA formulations (Figure 2). These results indicated that FLUVA preferentially suppresses the self-renewal and proliferation properties of BCSCs. In addition, the anti-sphere formation effect of HA-L-FLUVA was much stronger than that of FLUVA and L-FLVUA (Figure 3 and Table 2). These results indicated that FLUVA exerted an anticancer effect and that HA-conjugated liposomes can enhance the effect of FLUVA by targeting BCSCs. 

Finally, HA-L-FLUVA also showed a more distinctive in vivo anticancer effect than free FLUVA or DOX, as evidenced by the change in tumor size and survival rate in xenografted mice. The combination therapy with HA-L-FLUVA + DOX was the most effective among the five treatment groups, leading to more prolonged remission than that with the other drugs (Figure 4A). In addition, we investigated the anticancer effect and safety of various FLUVA formulations using Kaplan–Meier plots. All mice injected with FLUVA formulations were alive, although one death occurred in the group injected with HA-L-FLUVA. These results showed that there might be environmental factors other than the toxicity of the drug and liposome itself, and that it also has an anticancer effect. However, the death of mice in the group injected with DPBS and DOX is believed to be due to the increased cancer size and drug toxicity or other environmental factors (Figure 4B). Based on these results, FLUVA in the liposome formulation is likely to be protected from uptake by the RES, and the conjugation of HA to the surface of liposomes resulted in an increased stability and prolonged circulation in blood [32,41,52,53]. Based on the results shown in Figure 4A, HA-L-FLUVA probably showed the most potent anticancer effect and effective targeting of BCSCs. In addition, a loss of body weight of >10% after drug injection in mice is usually related to the toxicity of the drug or delivery system [46]. All groups of mice showed a slight weight loss of less than 10% after drug injection, but this was not statistically confirmed (Figure 6). Therefore, the FLUVA formulations were shown to exert therapeutic efficacy without causing toxicity as a treatment of BCSCs in animals. 

## 5. Conclusions

HA-L-FLUVA displayed much more potent anticancer effects than free FLUVA and DOX both in vitro and in vivo. HA served as a good targeting ligand for the CD44 surface marker of BCSCs, thereby allowing the specific delivery of FLUVA to BCSCs. Therefore, HA-L-FLUVA is expected to be useful as a novel therapeutic strategy for breast cancer therapy.

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
