# Peer review of "Repurposing of Fluvastatin as an Anticancer Agent against Breast Cancer Stem Cells via Encapsulation in a Hyaluronan-Conjugated Liposome"

_pharmaceutics, 2020, doi:10.3390/pharmaceutics12121133_

Round 1

Reviewer 1 Report

In present article Hyaluronan-conjugated fluvastatin-encapsulating liposomes  were evaluated for their anticancer efficacy in vitro and in vivo. The article is well written and easy to read and understand. The experiments are well done and propertly described. The article can be published after some corrections.

Corrections:

The problem of this article is partially unproper statistical analysis.

1) The data on animals should be analysed by non-parametic tests (Kruskal-Wallis ANOVA or something similar).

2) Figure 2. The statistical analysis should be made using ANOVA and post-hoc multiple comparisons tests (can be parametric).

3) Figure 4-A. It is difficilt to make statistical conclusions based on this figure. Please, duplicate the data in Table.

4) Fig. 7. There is no statistical confirmation of lost of body weight (10%). It means that authors can conclude: "body weight did not, practically, change". 

Reviewer 2 Report

In this manuscript hyaluronan-conjugated fluvastatin-encapsulating liposomes (HA-L-FLUVA) were evaluated for their anticancer efficacy in vitro and in vivo. The results indicate that HA-L-FLUVA could serve as an effective anticancer drug by inhibiting the growth of tumor in breast cancer stem cells (BCSCs) xenograft mouse models. As state in the comments below, the current data is not sufficient and needs major revision.

1. Is FLUVA encapsulated in the liposome membrane or aqueous core? The encapsulation efficiency is 33-35%. However, it is not clear from the preparation method how the unencapsulated FLUVA was removed.

2. The surface of the liposomes was modified with hyalurona to specifically target BCSCs via CD44 which is overexpressed on the cancer stem cells as shown in Scheme 1. It is obvious from Figure 2, Figure 3, and Table 2 that both FLUVA and its liposomal formulation are effective to inhibit the proliferation of BCSCs in dose-dependence manner. However, the data is insufficient to support that HA acts to target the CD44 in the present liposomes because the comparable data of L-FLUVA without HA is missing. Authors should add comparable data of L-FLUVA in Figure 2, Figure 3, and Table 2 to support the significance of HA to interact with CD44-positive cells.

3. In addition to the comment 2, the comparison of anti-proliferative effects of FLUVA, L-FLUVA, and HA-L-FLUVA for CD44-negative cancer cells provides further evidence to support the targeting of breast cancer stem cells via CD44 by hyalurona-conjugated liposomes.

4. It is not clear why DOX was used with HA-L-FLUVA. Some possible effects of the combination therapy due to combination of liposomal drug and free drug or combination of FLUVA and DOX as well as HA-L-FLUVA and DOX can be estimate. Is there any expect for specific synergy effect by the combination of HA-L-FLUVA and DOX? This point should be discussed.

5. Much amount of FLUVA and HA-L-FLUVA were administrated in the mice compared with that of DOX. Because the drug efficacy is dose-dependence in general, the DOX might be more effective than HA-L-FLUVA and FLUVA when the DOX is administrated with same dose with HA-L-FLUVA and FLUVA. While, the toxicity of drugs might be limited the dose. Authors should explain the comparison between 3 mg/kg DOX every 4 days and 15 mg/kg HA-L-FLUVA every day is reasonable.

6. In Figure 4A, the hypothesis for statistical analysis is not clear. Student’s t-test should not be correct to compare between many groups.

7. As authors mentioned, the presence of a small population of breast CSCs (BCSCs) is recognized as one of the causes of breast cancer recurrence. In this sense, the important point in research to target BCSCs can be regarded to prevent the recurrence. However, the present research is not reach to the evaluation of recurrence. Authors should address this point.

Round 2

Reviewer 2 Report

The authors have addressed all comments from the reviewer. The manuscript has been significantly improved and can be recommended for publication in Pharmaceutics.